# Correlating the Performance of a Fire-Retardant Coating across Different Scales of Testing

**DOI:** 10.3390/polym12102271

**Published:** 2020-10-02

**Authors:** Yan Hao Ng, Indraneel Suhas Zope, Aravind Dasari, Kang Hai Tan

**Affiliations:** 1School of Civil and Environmental Engineering (Blk. N1), Nanyang Technological University, 50 Nanyang Avenue, Singapore 639798, Singapore; yhng@ntu.edu.sg; 2School of Materials Science and Engineering (Blk. N4.1), Nanyang Technological University, 50 Nanyang Avenue, Singapore 639789, Singapore; iszope@ntu.edu.sg

**Keywords:** multi-scale, fire retardant coating, cone calorimeter, furnace tests, ISO 834, fire curves

## Abstract

Material-scale tests involving milligrams of samples are used to optimize fire-retardant coating formulations, but actual applications of these coatings require them to be assessed with structural-scale fire tests. This significant difference in the scale of testing (milligrams to kilograms of sample) raises many questions on the relations between the inherent flammability and thermal characteristics of the coating materials and their “performance” at the structural scale. Moreover, the expected “performance” requirements and the definition of “performance” varies at different scales. In this regard, the pathway is not established when designing and formulating fire-retardant coatings for structural steel sections or members. This manuscript explores the fundamental relationships across different scales of testing with the help of a fire-protective system based on acrylic resin with a typical combination of intumescent additives, *viz*. ammonium polyphosphate, pentaerythritol, and expandable graphite. One of the main outcomes of this work dictates that higher heat release rate values and larger amounts of material participating in the pyrolysis process per unit time will result in a rapid rise in steel substrate temperature. This information is very useful in the design and development of generic fire-retardant coatings.

## 1. Introduction

It is a regulatory requirement in most countries to protect structural steel members against fire. The common way of achieving this is by using coatings (such as intumescent and cementitious) and testing the protected steel members to meet certain fire resistance criteria (e.g., deflection limits and/or critical temperature) with the imposed mechanical and thermal loads according to standard fire tests such as BS476 Part 21, ASTM E119 or EN 13381-8. However, such large-scale fire tests are usually time-consuming, expensive, and require a lot of resources. In this context, many questions arise on the validity of material-scale and bulk-scale thermal and flammability tests as well as their usefulness in identifying the performance of coatings at the structural level without any further fire testing. However, unlike bulk-scale tests, at the structural scale, (protected) members are exposed to heating on all sides, and heat transfer occurs in all directions. Hence, the dimensions of the members and their section factors should also be considered.

At the material scale, Underwriter Laboratories (UL) 94, limiting oxygen index (LOI), thermogravimetric analysis (TGA), and pyrolysis combustion flow calorimetry (PCFC) are a few commonly used techniques to characterize parameters relating to thermal stability, flammability, and chemical processes occurring in flaming combustion (pyrolysis of the polymer and oxidation of the volatiles). TGA and PCFC data are based on milligrams of a specimen, whereas LOI and UL tests are limited to rectangular strips of 125 mm × 12.5 mm × (0.8 to 3.2) mm exposed to a small candle-like flame. In LOI, the minimum oxygen requirement for a specimen to sustain a flame is measured. In the vertical burning UL94 test, a flame source (with a 20 mm flame height) is applied to the bottom of the specimen for 10 s for two applications, and flammability (vertical upward burning and burning time) is measured along with a qualitative monitoring of its dripping behavior (melt and flame). As expected, the performance of the specimen is thickness dependent. This illustrates the fact that these small-scale “reaction to small-flame” tests are not pointing to intrinsic material properties. Therefore, extreme caution is required when correlating these data to other tests. 

PCFC provides information on the heat release rate and, in turn, the oxidative combustibility of volatiles generated during the pyrolysis of a polymer. However, mechanistic features such as dripping, swelling, shrinking, intumescent effect, or any other physical features of the decomposing sample are not considered in PCFC. This is also apparent from the amount of the specimen used in testing. This is the reason why Schartel et al. [1] proposed that “PCFC is constructed to be blind to flame inhibition.” This is also true to certain extent in a forced-combustion bulk-scale test such as a cone calorimeter, which provides heat release rate data along with other important information such as ignition time, smoke production rate, and total heat released [2,3]. However, even this forced flaming combustion test simulating a developing fire scenario cannot be used to determine the “true” combustibility of a specimen. Combustibility is determined by standard tests such as BS EN ISO 1182 or BS 476: Part 4. Although the cone calorimeter data is influenced by the development of a physical barrier during the combustion process, thus registering a broader heat release rate plot, this test does not give any information on the gas phase mechanisms. Orientation (horizontal or vertical) and sample confinement (restrained or unrestrained) are other parameters that have to be considered in this test. 

Nonetheless, there are some studies correlating the behavior of samples observed in small-scale tests such as PCFC with UL 94 and LOI tests with cone calorimeter tests [4,5,6,7,8]. Figure 1 summarizes, schematically, the concept behind the different tests and highlights some of the correlations that have been noted among the different tests in the literature. For example, it was reported that polymeric materials that showed an LOI around 30 vol% and above exhibit a V0 rating in UL 94 test. However, as briefly discussed earlier, correlating across different testing standards is not straightforward, as they are not monitoring the intrinsic material properties. In addition, the “demanding” versus “less demanding” configuration of specimens in these tests—that is, “upward” (in UL 94) versus “downward” (in LOI) burning—suggests that the correlations observed should be “not accurate” and cannot be generalized or accurately correlated with other fire tests. Schartel and Hull [9] reported that despite not scientifically proven, if the averaged heat release rate (HRR) is in the range of 80 to 300 kWm^−2^ at low applied heat fluxes (25 to 35 kWm^−2^), then a good correlation could be achieved with UL 94 and LOI results.

Apart from the above, there are publications relating the data from different polymers with rapid mass calorimeter, cone calorimeter, and PCFC [11,12]. More importantly, from the context of this manuscript, there are also many publications on different fire-retardant polymer-based coatings on small steel panels [13,14,15,16], some [17,18,19,20,21,22,23,24] of which use a bulk-scale test setup to evaluate the performance of intumescent coatings. Bulk-scale test involves the use of a coated substrate and the amount of materials involved is typically measured in grams. More information on the different scales can be seen in Table 1. Jimenez, Duquesne, and Bourbigot [19] used a heat radiator setup and correlated the performance (time to critical temperature) of intumescent coatings on steel plates tested in an industrial furnace. However, it is worth noting that the specimens were exposed to heat in a one-dimensional manner. There are publications that proposed experimental procedures or setup to simulate heat flux [25] or gas pressure [26] in a structural-scale furnace, but certain limitations apply with the reduction in scale, and the focus has largely been on the temperature development of the substrate. Correlating numerical simulations with cone calorimeter results [18,27,28] and furnace test results [29] obtained at the bulk scale are also proposed, but beyond bulk-scale tests, there seems to be little information in the literature [30,31]. Even so, we consider the following questions. 

Is it reasonable to provide suggestions/insights based on the data from material- and bulk-scale tests on how a fire-protective coating formulation behaves when tested at the structural scale?Is there any evidence to correlate the results of a coating from a laboratory-type furnace in a one-dimensional setup with the behavior in structural-scale fire tests where all sides of the coated member are exposed to the fire curve?

These critical questions form the basis of the manuscript, which aim to understand the implications of the test data obtained at different scales. Ultimately, the objective is to identify any qualitative relation that provides an early indication of the performance of the coating at the structural scale. This information can be used to screen potential formulations before testing them in a structural-scale fire test, which is expensive. For this purpose, a conventional coating formulation consisting of ammonia polyphosphate, pentaerythritol, and expandable graphite in an acrylic resin was chosen. The different thermal, flammability, and fire tests across scales that were chosen to be a part of this work include TGA, PCFC, cone calorimeter, 1D heat transfer in a laboratory furnace, and structural-scale fire tests where the I-section is exposed to heat from all sides. Extrinsic factors such as the section factors, edge effects of steel columns, etc. are not considered in this study. 

## 2. Experimental Section

### 2.1. Materials

Ammonia polyphosphate (APP), pentaerythritol (PER), and expandable graphite (EG) were used as intumescent additives in the ratio of 3:1:1 at a total loading level of about 41% in a water-based acrylic resin. The additives and resin were obtained from commercial sources; acrylic resin was supplied by Synthomer PLC (London, UK).; APP (phase II) and PER were obtained from McKinn International Pte Ltd. (Singapore); and EG was supplied by Shijiazhuang ADT Carbonic Material Factory (Shijiazhuang, China). 

### 2.2. Characterization and Fire Tests

The model coating system was assessed using five types of tests (i.e. TGA, PCFC, cone calorimeter, furnace tests, and structural-scale fire test) conducted at three different scales (i.e., material, bulk, and structural). The three different scales are determined based on the amount of sample involved in the test. A summary of the test parameters for the five types of thermal/flame/furnace tests is shown in Table 1. The coatings were sprayed over the steel substrate, layer by layer, until the desired dry film thickness was achieved. The dry film thicknesses of the coatings were measured with an Elcometer 456 (Manchester, UK), which uses the electromagnetic induction principle to measure the thickness. The average dry film thicknesses presented in Table 1 deviates slightly from the targeted dry film thicknesses of 2.5 and 5 mm. For ease of comparison between the bulk- and structural-scale tests, the coatings will be referred to as 2.5 and 5 mm instead of the average dry film thicknesses. It is apparent in Table 1 that the heating rates also differ with different scales of testing. Further, material-scale tests only involve the fire-retardant coating without a substrate.

### 2.3. Thermogravimetric Analysis

The mass loss rate of the sample with temperature was measured using a TA Instruments SDT Q600 (New Castle, USA). The samples, weighing 8 to 10 mg, were placed in a sample holder (e.g., alumina cup or platinum pan) and heated up to 900 °C with a heating rate of 0.33 K/s. Air or nitrogen was supplied into the chamber to simulate an oxidative or pyrolysis environment, respectively. The tests were repeated once to ensure repeatability of data.

### 2.4. Pyrolysis Combustion Flow Calorimetry (PCFC)

PCFC measurements were taken using an FTT FAA Micro Calorimeter (East Grinstead, UK) according to Method A, as it closely represents the mechanistic behavior of a coating in a furnace (flaming combustion), characterizing the sequential steps of pyrolysis of the coating (through thickness) and thermal oxidation of the evolved volatiles. The samples, weighing 6 to 17 mg, were placed in an alumina cup and heated up to 600 °C in the pyrolysis chamber with a heating rate of 0.33 or 1 K/s. The combustion chamber was kept at 900 °C to ensure all volatiles were combusted and quantified. The data presented in this manuscript are based on an average of 2 to 3 samples. The sample mass was adjusted to ensure that the minimum oxygen concentration level reduces to 10 ± 3%, as recommended by the manufacturer. However, the test involving the acrylic resin with an intumescent formulation at 0.33 K/s has a minimum oxygen concentration level of 16%. To meet the required minimum oxygen concentration level of 10 ± 3%, it is expected that the mass of the sample will need to be increased to at least 30 mg. This will result in the release of excessive volume of volatiles and will result in inaccurate results. Hence, only for the test involving acrylic resin with intumescent formation at 0.33 K/s, the minimum oxygen concentration level is not used as a validity check. 

### 2.5. Cone Calorimeter 

The combustibility and heat release rate of the samples were assessed with an FTT cone calorimeter (East Grinstead, UK). Mild steel plates with dimensions of 90 mm by 90 mm by 4 mm were spray-coated to the desired thickness. There were 2 different series of cone calorimeter tests conducted, and the results were compared within each series to minimize calibration or instrumentation errors. For the first series, an irradiance flux of 50 kW/m^2^ was imposed onto the coating surface of the samples, and the test was stopped when the HRR was stabilized. In the second series of tests, irradiances of 35 and 50 kW/m^2^ were used, but the test was stopped after 1 h of heat exposure. The steel substrate temperature was continuously measured with K-type thermocouples embedded onto the steel plate and connected to a data-logger. Despite the expected changes in heat flux experienced by the samples, expansion of the coating is able to simulate the more realistic temperature rise of the steel substrate. A test in one of the series was repeated to ensure consistency in data.

### 2.6. Bulk-Scale Furnace Tests 

The bulk-scale furnace tests were carried out using an electric furnace of dimensions 0.7 m wide by 0.85 m deep by 1.7 m long, which is capable of closely simulating the ISO 834 standard fire curve (Figure 2a). The coated steel plates were encased with gypsum boards (thickness of 15 mm) to restrict heat transfer to a single face (one-directional heat transfer) (Figure 2b) [32]. This provided a controlled setup that was convenient to prepare for bulk-scale tests of coated steel samples. Fire-retardant coatings with two different dry film thicknesses (≈2.5 mm and ≈5 mm) were studied. A type-K thermocouple was used to record the temperature development of coating–steel interface (Figure 2c). The result of one bulk-scale furnace test is discussed in this manuscript.

### 2.7. Structural-Scale Fire Test

The structural-scale fire tests were performed using a gas-fueled furnace 4 m long by 3 m wide by 2.2 m deep, which is capable of closely simulating the ISO 834 standard fire curve. The key test parameters were carried out with reference to EN 13381-8. Unlike the bulk-scale furnace tests, the samples were placed within the furnace and were exposed to heating from all sides (Figure 3b). Hence, the steel section, UB 406 × 178 × 67, was coated all around and has a section factor of 175 m^−1^. There were four type-K thermocouples to record the temperature of both flanges (two thermocouples for each flange) and one type-K thermocouple to record the web temperature. The result is based on the average temperature of the thermocouples obtained from the steel section with a dry film thickness of ≈5 mm. The result of one structural-scale fire test is discussed in this manuscript.

## 3. Results and Discussion

As the FR mechanism of similar intumescent coating systems has been well documented in the literature [33,34,35,36], this manuscript will not focus on its FR mechanism. Instead, the emphasis will be on identifying the key performance pointers in the material- and bulk-scale tests that could provide an indication of how the coating might perform in the structural-scale fire tests. 

### 3.1. TGA and PCFC Data and their Correlation with the Imposed ISO 834 Fire Curve 

The HRR profiles, obtained from PCFC tests at 0.33 and 1 K/s, for neat resin (ACR) and its intumescent formulation (ACR-FR) are shown in Figure 4a. As expected, an increase in the peak HRR and a slight shift of the temperature corresponding to peak HRR toward higher temperature were noted with an increase in the heating rate. ACR-FR showed an increase of peak HRR from 81 to 219 W/g when the heating rate was increased from 0.33 to 1 K/s. The corresponding temperature increased from 366 to 408 °C. The same was also true for neat resin (166 W/g at 0.33 K/s to 464 W/g at 1 K/s, with the corresponding temperature increased from 383 to 409 °C, respectively). Therefore, these changes could not be attributed to the inherent chemical or physical nature of the materials. Similar behavior was also observed in many other systems [5,7,37,38]. This suggests that there is a direct correlation between the heating rate and the rate of pyrolytic decomposition (and simultaneous oxidation).

The heat release capacity ($\eta_{c}$) is a function of the heat of combustion of fuel gases per unit initial mass of solid ($h_{c,s}^{0}$), global activation energy for pyrolysis ($E_{a}$), temperature of maximum pyrolysis (mass loss) rate ($T_{p}$), natural number (*e*), and gas constant (*R*) [38,39], that is,
(1)$$\eta_{c}=\frac{h_{c,s}^{0}E_{a}}{eRT_{p}^{2}}$$

The HRC ($\eta_{c}$), as shown in Equation (1), is viewed to be a time- and rate-independent material property [39]. In line with this, in the current study, there were minimal differences in the HRC of the ACR-FR system, ≈249 J/(gK) at 0.33 K/s compared to ≈243 J/(gK) at 1 K/s. Even for neat resin, variations in the HRC values were minimal, *viz.* 521 J/(gK) at 0.33 K/s to 517 J/(gK) at 1 K/s. In fact, Lyon and Walters [40], after evaluating different polymer systems in a microscale combustion calorimeter, proposed a linear relation between the maximum specific heat release rate and the heating rate until ≈5 K/s. Beyond 5 K/s, the linear relation was no longer valid due to a significant thermal lag between the sample and the quartz tube that held the sample for a higher heating rate than 5 K/s. 

However, it was interesting to note that despite the higher peak HRR of the neat acrylic system, irrespective of the heating rate, the heat of complete combustion (HCC) values suggested otherwise. HCC is derived based on the ratio of total heat release and char yield. HCC values for ACR-FR were 19.5 kJ/g at 0.33 K/s and 18.9 kJ/g at 1 K/s, while for ACR, they were 16.2 kJ/g at 0.33 K/s and 15.5 kJ/g at 1 K/s (normalized with respect to the resin content). Although small, the differences might suggest a change in the volatiles released during the pyrolysis of the intumescent formulation compared to the neat polymer resin. This might possibly be due to secondary reactions between APP and acrylic, apart from the primary intumescent reaction between PER and APP. Nonetheless, the temperature of the peak HRR obtained from PCFC closely matches with the peak decomposition temperature in TGA (both in nitrogen and oxygen environments) (Figure 4b). This was irrespective of the heating rates within each TGA or PCFC test. This suggests that the mass pyrolyzed was combustible in the temperature range of 300 to 400 °C. If there were to be wide differences in the temperature range of derivative mass loss in TGA and the temperature of peak heat release rate in PCFC curves, it could suggest that the mass lost in TGA did not lead to combustible volatiles, which is not the case here. Oxidation of the char, as identified in TGA in the temperature range of 500 to 600 °C in the air environment, understandably, would not be possible in PCFC. Another similarity is that the residual masses of the samples for these two tests were similar. The residual mass in PCFC for the ACR-FR system when conducted at 0.33 and 1 K/s was 23% and 20%, respectively, while in TGA in air and nitrogen at 600 °C, it was 20% and 22%, respectively.

As gleaned from the PCFC and TGA results (Figure 4a,b), and correlating the findings to an imposed ISO 834 fire curve (Figure 5), the following findings can be broadly suggested:The high heating rates expected in the ISO 834 fire curve scenario, particularly during the initial 15 min (Table 2), suggest that a higher value of peak HRR is to be expected during the initial period. This can determine the initial rise of temperature of the steel substrate underneath, depending on the volume (thickness) of the coating material that is involved in the pyrolysis during this period. It is important to note that in TGA and PCFC tests, only a specific heating rate was employed throughout the test, while in the furnace test, the imposed heating rate (K/s) varied widely throughout the test. This aspect in combination with the intumescent phenomenon should be considered while correlating the data among the tests. Therefore, the rise in temperature of the steel substrate with time is taken as an important parameter while proposing the relationships among the different tests.The values of peak HRR, THR, and HCC are critical parameters that can affect the continuous flaming of the sample, ultimately influencing the heat transfer to the steel substrate.Thermal stability variations of polymers (even with an increment of 30 to 50 °C from the base system) possibly will have a negligible effect considering the high heating rate for the ISO 834 fire test and the achievement of ≈683 °C (in the first 10 min), which is well beyond the thermal decomposition temperature of most of the commonly used polymers.

### 3.2. Correlating the Data from TGA, PCFC, and Cone Calorimeter 

When changing from material-scale to bulk-scale, the thickness of the sample, the mechanisms of degradation, and the type of additives become more critical. This was obvious from a comparison of PCFC and TGA results (Figure 4a,b) with cone calorimeter data in terms of mass loss and heat release rates, as shown in Figure 6.

The sharp loss in mass, typically seen in TGA between 300 and 400 °C for both neat acrylic resin and its intumescent formulation, or the rapid increase in HRR seen in PCFC in similar temperature regimes (involving the entire volume of the sample) was not so obvious in the cone calorimeter tests. An example of this behavior for a 2.5 mm thick coating is shown in Figure 6. This did not mean that there was no rise in the temperature of the substrate or HRR. Considering the imposed heat flux of 50 kW/m^2^ on the sample in the cone calorimeter test (that is, a rapid heating rate or higher temperature gradient across the thickness of the sample), this would result in pyrolyzing a huge volume of the sample in the first few minutes of the test, while a homogeneous insulating barrier beneath the pyrolysis zone was simultaneously established. The insulating barrier formed on the surface of the sample was due to the expansion of EG, which reduced the heat transfer to the underlying layers. It should also be noted that due to swelling of the sample, the heat flux imposed on the sample surface in the cone calorimeter test would be different from the initial heat flux. Thus, the thermal load varied during the test. This is an important parameter that could not be considered in the current work. Nonetheless, it is important to note that the above behavior depends on the thickness of the sample. That is, for thermally thin samples, the entire volume of the sample would be pyrolyzed simultaneously due to the absence of thermal or viscosity gradients through the thickness of the sample. In this case, the heat release rate profile might be very similar to PCFC, and thus, the peak HRR depends on the (specimen’s contribution to the) total fire load. In the current scenario, thermally thick samples (2.5 and 5 mm) result in a broadening of HRR curves. This is also evident in Figure 6, in which the decreasing gradient of mass loss with increasing time is observed: −0.156 %/s from 0 to 400 s, −0.0792 %/s from 400 to 600 s and −0.0422 %/s from 600 to 1100 s.

Depending on the volume of sample pyrolyzed per unit time, the rise in substrate (steel) temperature would vary. The measured temperature profiles of the steel substrate in a cone calorimeter along with the HRR patterns shown in Figure 7 confirm the relationship between pyrolysis and rise in temperature of the steel substrate.

As evident, by 200 s, the steel temperature in the cone calorimeter test was raised to ≈178 and ≈98 °C for 2.5 and 5 mm thick coatings, respectively. For example, within that 200 s, there is almost 35% mass loss for the 2.5 mm thick sample, which also explained why the time-to-ignition of the ACR-FR system is ≈9 s for 2.5 mm. Furthermore, the time-to-flame out (≈401 and ≈848 s for 2.5 and 5 mm, respectively) differed for the two samples, confirming that pyrolysis was responsible for the initial rise in temperature of the steel substrate. 

In summary, many similarities and critical information could be drawn from the material- (PCFC/TGA) and bulk-scale tests such as cone calorimeter. The thickness of the sample (coating), imposed heating rate, and the volume of flammable/combustible volatiles released are a few key parameters that provide valuable insight on how the coating might behave in a structural-scale test where an ISO 834 fire curve is imposed. Before discussing this further, it should be noted that the heat flux in a furnace test will vary with thermal load, which differs from a cone calorimeter test where heat flux is maintained at a constant rate throughout the test. 

### 3.3. Correlating the Data from Cone Calorimeter, Bulk- and Structural-Scale Furnace Tests

Interestingly, the above observations on the increase in temperature of the steel substrate due to pyrolysis also hold for a 1D heat transfer (bulk) test. As the heating rates in the first 10 to 15 min in the ISO 834 fire curve are on par with the imposed heating rates in PCFC (Table 2) or higher temperature gradient experienced in the cone calorimeter, a higher value of peak HRR is expected in the furnace test during the initial period. This is due to the higher volume (thickness) of material involvement in the pyrolysis. This results in ignition/flaming of the sample until the HRR values are lowered due to the formation of an insulating barrier. As a result of this, the initial steep gradient in the derivative time–temperature profile of the underneath substrate is noted in Figure 8. With 2.5 mm thick coating, at 15 min, the steel temperature was 214 °C, and with 5 mm thick coating, it was about 149 °C. As shown in Figure 7a,b, a similar trend was observed even with the temperature profiles obtained from cone calorimeter tests. 

After the matrix resin has been completely pyrolyzed, the gradient of the time–temperature profile (furnace test) would only depend on the thermal conductivity of the char, and so the gradient would be constant, as the thermal load in the furnace was almost constant during this period. For example, a constant gradient of 0.0767 K/s was observed for a 2.5 mm thick sample between 15 and 77 min, while 0.0675 K/s was noted for a 5 mm thick sample between 18 and 97 min. The thermal conductivity value of the char has a significant influence on the performance of the coating, as it governs the rise in temperature over a much longer period of time. Preliminary studies (that have not been published) on the ACR-FR suggested that effective conductivity after the average coating temperature reaches more than 300 °C is relatively constant. Average temperature of the coating is estimated based on the temperature of the surface and the bottom of the coating. The surface temperature is calculated based on heat balance equation with the irradiance flux of the conical heater while the bottom temperature is measured using a type-K thermocouple. The conductivity of the coating changed from 0.441 W/(mK) at room temperature to 0.082 W/(mK) at 300 °C and stabilized around 0.03 W/(mK) at higher temperatures above 600 °C. With this parameter, it is possible to predict the rise in temperature of the substrate.

In Figure 9, the time–temperature profiles for the steel substrate obtained in the cone calorimeter, bulk- and structural-scale tests are presented. Generally, a developing fire is characterized by an external heat flux of 20 to 60 kW/m^2^, while a fully developed fire is characterized by an external heat flux much greater than 60 kW/m^2^ [9]. Cone calorimeter data obtained with heat fluxes of 35 and 50 kW/m^2^ are shown in Figure 9. It is interesting to note that the time–temperature profiles obtained with cone calorimeter (at both 35 and 50 kW/m^2^) showed a parabolic nature with decreasing gradient and attained almost a plateau in temperature after 35 min into the test. The swelling of the sample and the formation of an expanded graphite network resulted in a change of irradiance imposed on the sample. As the oxidation of this network seemed to be relatively absent under the imposed conditions, it caused a constant temperature profile after 35 min. However, in the actual furnace tests (both electric and gas-fueled), the plateau was not noticeable due to relatively higher temperatures and possibly higher flux imposed on the sample. Therefore, the behavior of structural-scale tests deviated from the heat release rate plots obtained in cone calorimeter tests. Other factors such as the thickness and extent of swelling might further influence the differences while comparing cone calorimeter data with furnace data.

The complications will further increase if the discussions and observations are extended to full-scale testing of a column ranging from 1 to 4 m length and with different section factors. Some of the extrinsic issues that can be expected include: higher heat fluxes imposed to the edges of an H or I steel section, irregular coverage of expanded char on those edges (depending on the section factor) resulting in through cracks, cohesion of char across the surface areas, self-weight of char, and more importantly, the mechanical load imposed on the column during the fire test. In fact, preliminary investigations on ACR-FR even under non-load-bearing conditions have revealed that the cohesion of char was poor due to the rapid and non-directional expansion of EG. More importantly, delamination between the char and the substrate occurred easily due to a reduction in the contact surface area as a result of expansion of EG/char. These parameters resulted in poor performance, which was observed in Figure 9, as the temperature of steel was obviously much lower in the 1D bulk-scale test compared to the structural-scale test after 30 min. Hence, the negative performance of the coatings at structural scale was completely masked in 1D furnace tests.

### 3.4. ‘Indicators’ Identified for Materials- to Structural-Scale Tests

To summarize the major points that were deduced across the scale of observations, a schematic temperature–time profile of an FR coating on a steel substrate was prepared and shown in Figure 10. Three different segments were highlighted in the profile based on the observations presented in earlier sections, and the importance of those segments and the key parameters governing those segments are elaborated further in the table associated with Figure 10. 

## 4. Conclusions

This manuscript dealt with identifying the key performance indicators in the material (TGA and PCFC)- and bulk (cone calorimeter and 1D furnace test)-scale tests of an FR coating that could assist in predicting/understanding its fire performance in the structural-scale fire test. The observations derived from these different tests at the structural scale are valid within the boundaries of the assumption that extrinsic parameters’ influence on the fire performance is negligible. In conclusion, the following relationships have been identified:Thermal and flammability characteristics of the coating obtained using TGA, PCFC, and cone calorimeter will affect the initial rise in steel temperature for the first 10 to 15 min of the bulk- and structural-scale fire tests.As the insulating barrier is formed on the top layers, the rate of pyrolysis of the underlying materials reduces and results in a slower increase in temperature of the steel substrate during the structural-scale fire test. Material-scale tests no longer provide much indication of the performance of the coating at this stage.Lastly, as the entire flammable content of the system is consumed, the leftover char’s conductivity governs the rate of increase in temperature of the steel substrate. It is expected to remain constant unless adhesion or cohesion failure of the char occurs. At this stage, extrinsic parameters dominate/control the rise in temperature of the substrate.

## Figures and Tables

**Figure 1 polymers-12-02271-f001:**
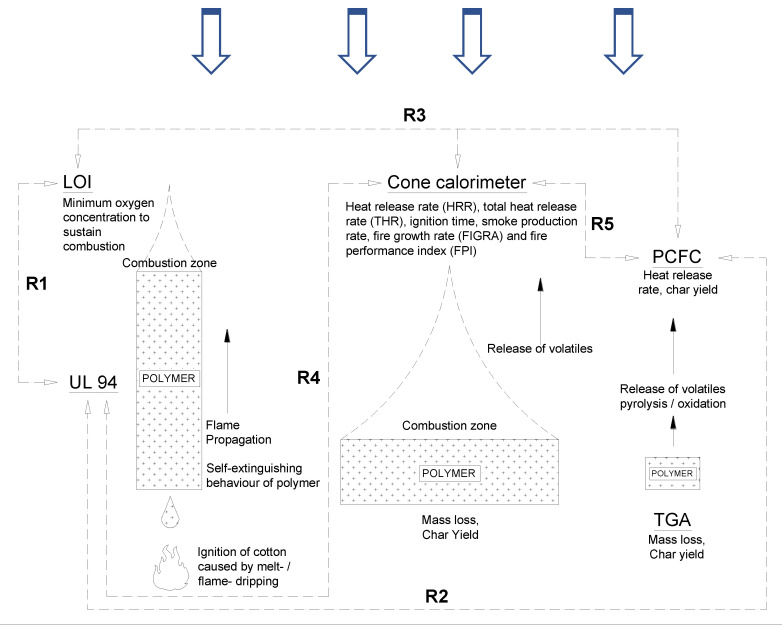
Concepts behind Underwriter Laboratories (UL) 94, limiting oxygen index (LOI), cone calorimeter, thermogravimetric analysis (TGA) and pyrolysis combustion flow calorimetry (PCFC). Some correlations observed in the literature between different tests are also shown [4,10].

**Figure 2 polymers-12-02271-f002:**
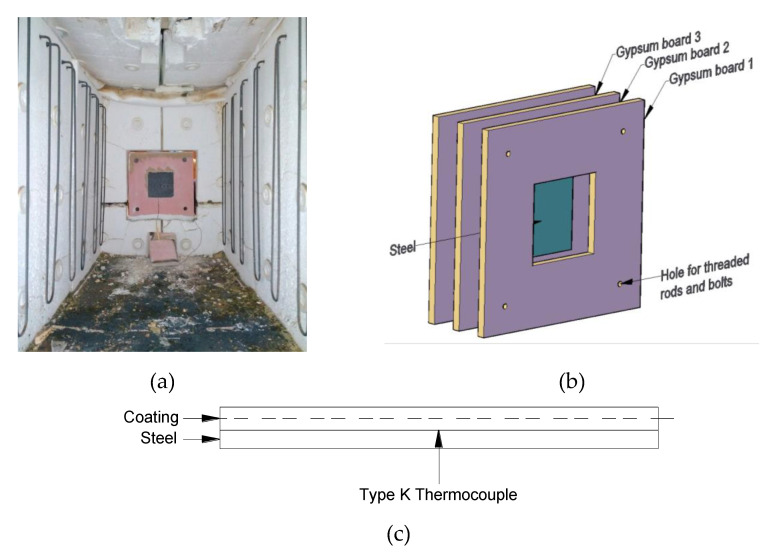
(**a**) Photo of the setup in the electrical heating furnace, (**b**) enlarged drawing of the setup, and (**c**) schematic drawing showing the location of the thermocouple.

**Figure 3 polymers-12-02271-f003:**
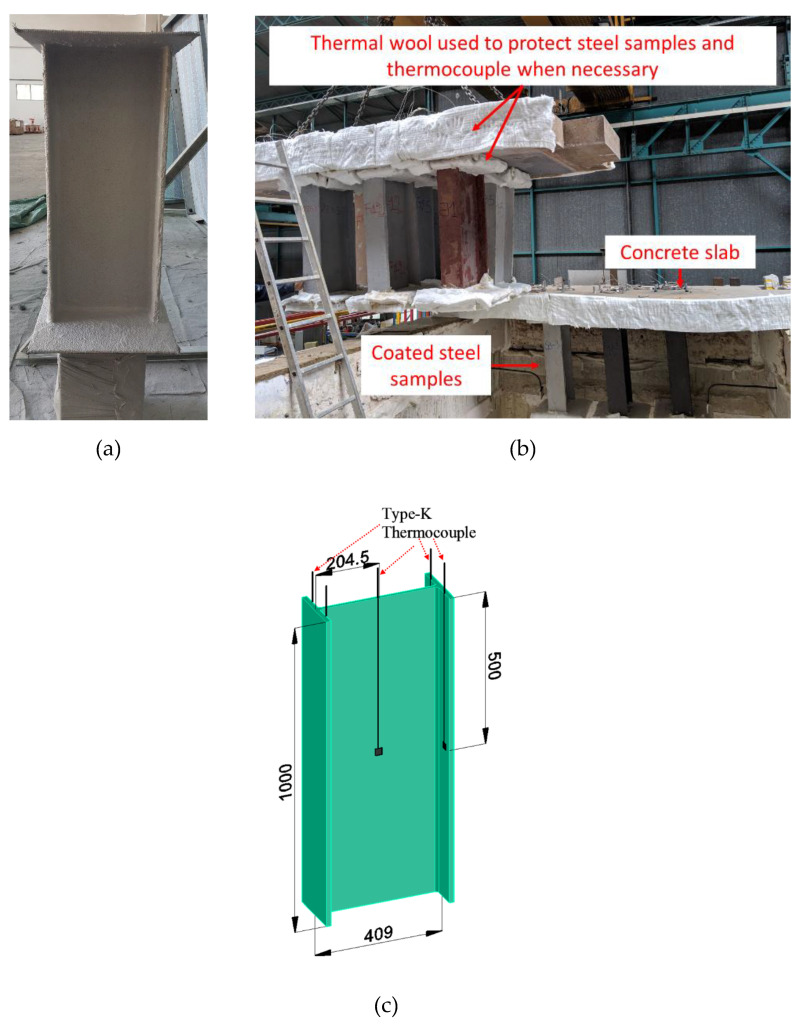
(**a**) Photo of a protected steel section and (**b**) typical furnace test setup for structural-scale members. (**c**) Schematic drawing showing the locations of the thermocouple.

**Figure 4 polymers-12-02271-f004:**
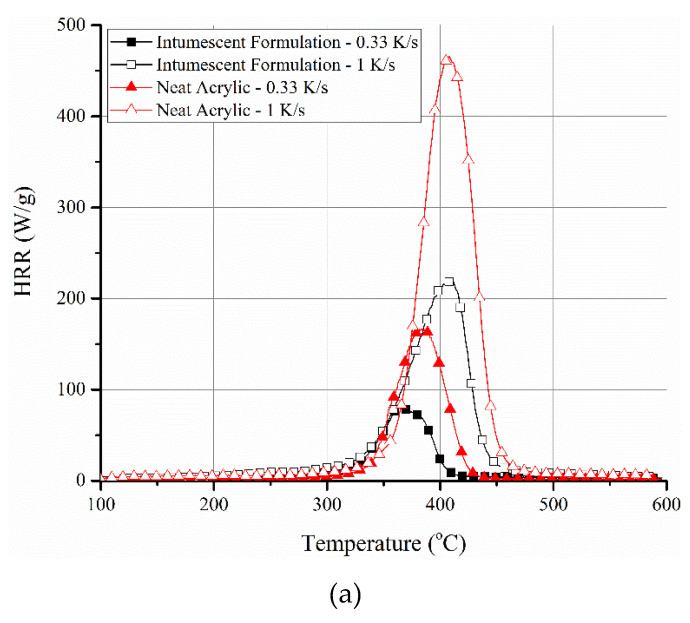
(**a**) PCFC data of neat acrylic resin and its intumescent formulation at different heating rates. A shift toward higher peak temperatures and corresponding higher peak heat release rate (HRR) values is evident with the increase in heating rate irrespective of the system. (**b**) TGA mass loss and derivative mass loss profiles of neat resin (ACR) and its intumescent formulation (ACR-FR) in air and nitrogen environments confirming close matching of peak decomposition temperature with peak HRR of ACR-FR in PCFC.

**Figure 5 polymers-12-02271-f005:**
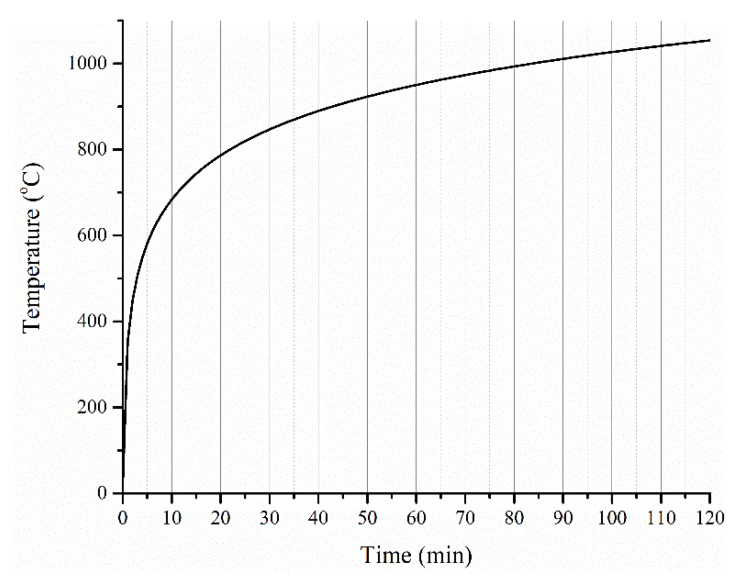
Standard ISO 834 cellulosic fire curve.

**Figure 6 polymers-12-02271-f006:**
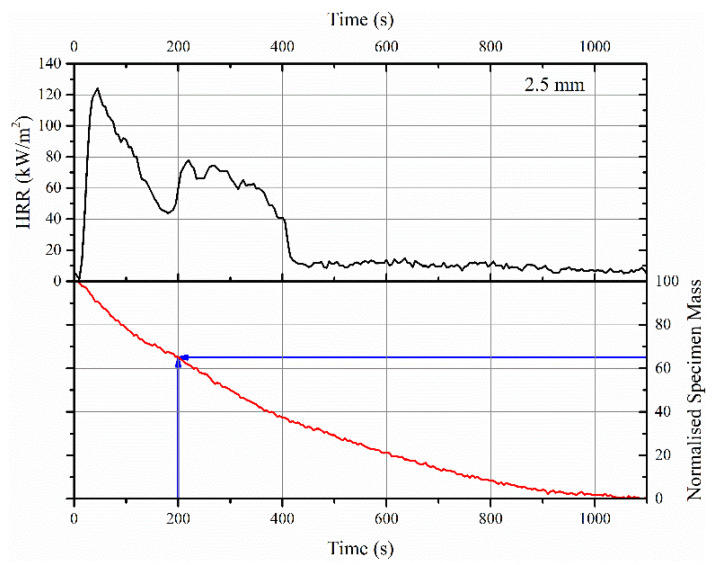
HRR curve and normalized mass loss profile obtained in cone calorimeter test for a 2.5 mm thick FR intumescent coating with an imposed heat flux of 50 kW/m^2^.

**Figure 7 polymers-12-02271-f007:**
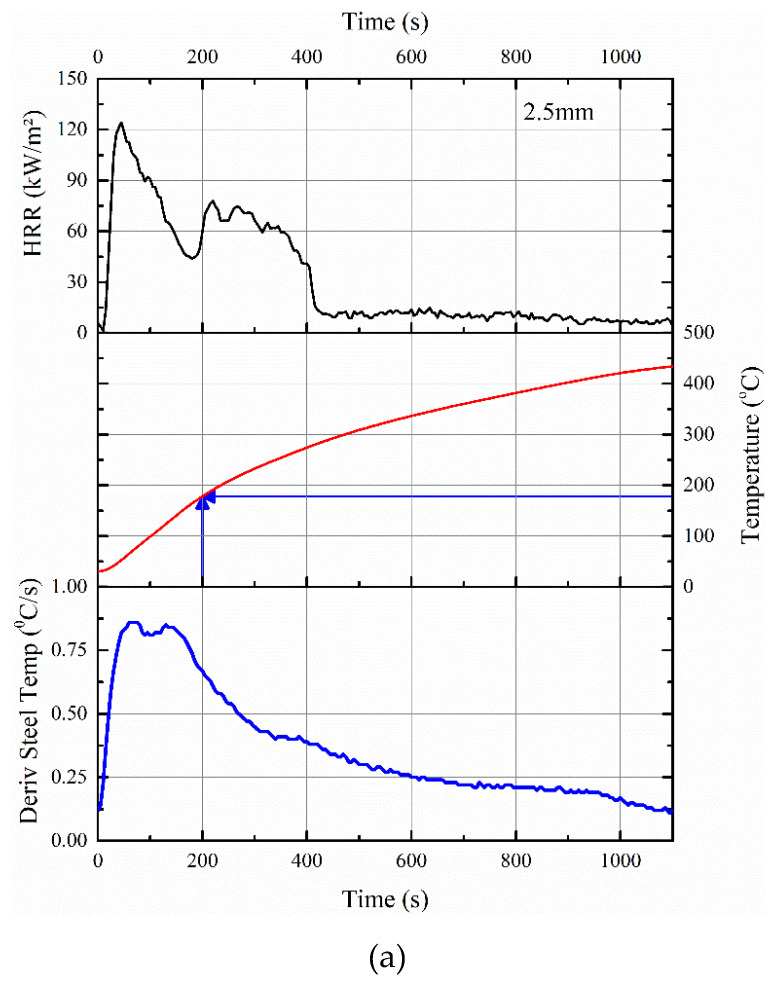
Correlations between HRR data in cone calorimeter, the rise in temperature of the steel substrate, and its derivative for (**a**) 2.5 mm thick and (**b**) 5 mm thick intumescent coating.

**Figure 8 polymers-12-02271-f008:**
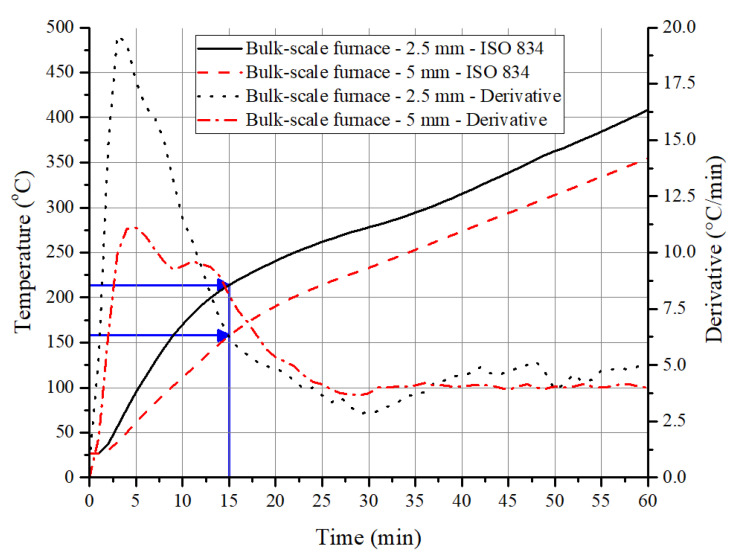
Time–temperature profiles and derivative curves obtained from steel substrate with 2.5 and 5 mm intumescent coating when exposed to ISO 834 fire curve in an electrical furnace.

**Figure 9 polymers-12-02271-f009:**
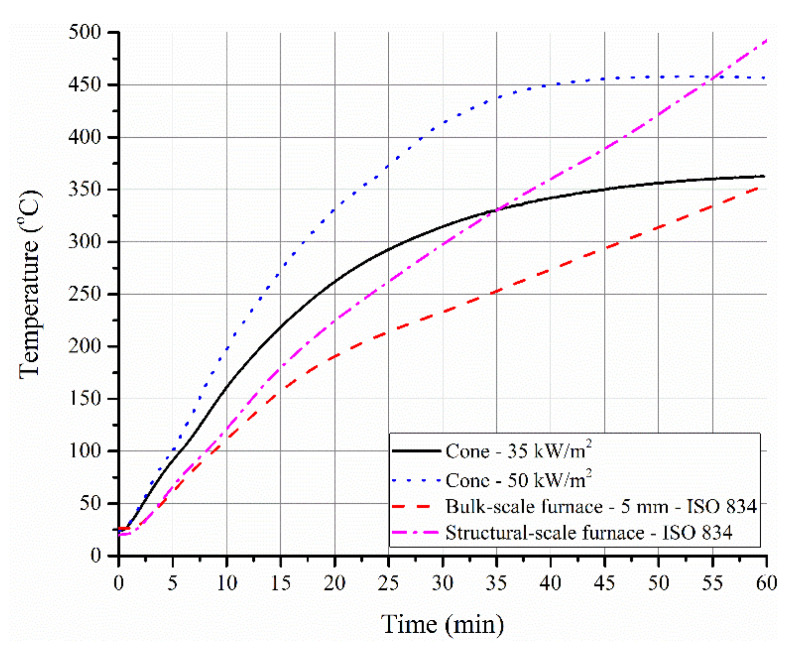
Time–temperature profiles of steel substrate in a cone calorimeter test conducted at 35 and 50 kW/m^2^ along with the profile of steel substrate with 5 mm thick coating in a furnace test.

**Figure 10 polymers-12-02271-f010:**
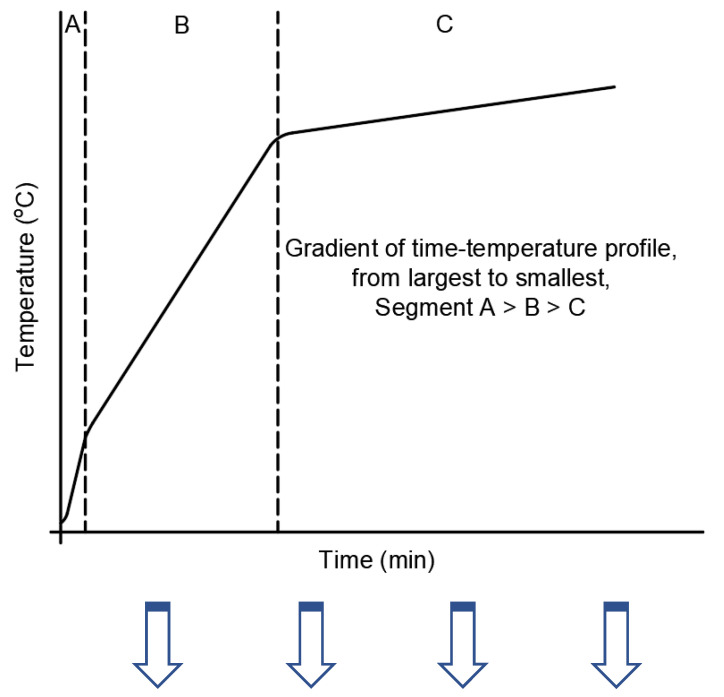
A schematic of the time–temperature profile highlighting the critical parameters that could be deduced from various tests across different scales. In segments B and C, the conductivity of the residue plays a critical role in determining the rise of steel temperature. Tests to measure high-temperature thermal conductivity should be included as well.

**Table 1 polymers-12-02271-t001:** Summary of test parameters at different scales.

Scale	Types of Test	Sample Size	Average Dry Film Thickness	Substrate Size	Heating Regime	Heat Source	Decomposition Atmosphere
Material	TGA	8 to 10 mg	N.A.	N.A.	0.33 K/s	Electric furnace	Air and nitrogen
	PCFC	6 to 17 mg	N.A.		0.33 and 1 K/s	Electric furnace	Nitrogen (pyrolysis chamber) and air (combustion chamber)
Bulk	Cone calorimeter with substrate temperature measurements	Coated steel plate	2.65 mm and 5.25 mm	90 mm × 90 mm × 4 mm	35 and 50 kW/m^2^	Radiant heating	Air
	Heat transfer tests	Coated steel plate	2.71 mm and 5.07 mm	160 mm × 160 mm × 0.6 mm	ISO 834	Electric furnace	Air
Structural	Heat transfer test	Coated steel column	5.2 mm	UB 406 × 178 × 67, with a length of 1 m	ISO 834	Gas- fueled	Air

Note: N.A. indicate that this parameter is not applicable to the test. Heating regime ISO 834 is described in Figure 5.

**Table 2 polymers-12-02271-t002:** Heating rates comparison among the different tests.

ISO 834 Standard Fire Curve		PCFC	TGA
Time Duration	Heating Rate (K/s)	Constant Heating Rate throughout the Test (K/s)	
Between 0 and 1 min	5.47	0.33 K/s and 1 K/s (imposed in this study)	0.33 K/s (imposed in this study)
Between 1 and 2 min	1.59		
Between 2 and 5 min	0.73		
Between 5 and 10 min	0.34		
Between 10 and 15 min	0.20		

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
