# Peer review of "Correlating the Performance of a Fire-Retardant Coating across Different Scales of Testing"

_polymers, 2020, doi:10.3390/polym12102271_

Round 1

Reviewer 1 Report

The authors describe how a small scale fire test can be correlated to larger scale testing. The research is very unique and interesting. The paper is very well written. Only the automatic numbering of figures and table went wrong. Many times ’error reference not found’ appears in the paper. Please check this before uploading the final document.

P8, top. The increase in peak maximum is at least in part easy to explain. The same occurs in a DSC experiment if you double the mass for instance (onset stays the same but peak shifts). Here off course, the mass of the sample stays the same but by scanning faster, the sample is brought to a higher temperature, thus according to Arrhenius, it will be faster. I don’t understand why at the end of the sentence, after pyrolytic decomposition (and simultaneous oxidation) is written. Also mention if exo is up or down when showing calorimetry results.

In conclusions 2nd bullet point

  • …Materials-scale tests no longer provides … write ‘provide’ rather than ‘provides’

Reviewer 2 Report

1.“PCFC provides information on the heat release rate and in turn, oxidative combustibility of volatiles generated during pyrolysis of a polymer. However, mechanistic features such as dripping, swelling, shrinking, intumescent effect, or any other physical features of the decomposing sample are not considered in PCFC.” How to present or investigate the dripping, swelling, shrinking, intumescent effect of coatings effectively?

2.”, thus registering a broader heat release rate plot, this test does not give any information on the gas phase mechanisms.”  The information on the gas phase mechanisms is interesting, but it is not explained in the article ?

3. “The samples, weighing 6 to 17 mg, were placed in an alumina cup and heated up to 600 oC in the pyrolysis chamber with a heating rate of 0.33 or 1 K/s. The combustion chamber was kept at 900 oC to ensure all volatiles were combusted and quantified.” It is better to clarify the difference between the CC and PCFC using the testing data!why the difference?

4. “The result is based on the average temperature of the thermocouples obtained from the steel section with a dry film thickness of ~5 mm.” what is the testing standard? Why is the 5 mm?

5 . The references are very old for readers and publications, you should cite more new references on the flame-retarding coating?

6. Why is the difference between the Thermogravimetric analysis, Pyrolysis combustion flow calorimetry (PCFC), Cone calorimeter, Bulk-scale furnace tests, Structural-scale fire test, please provides the reasons in details.
